# Holoprosencephaly: Review of Embryology, Clinical Phenotypes, Etiology and Management

**DOI:** 10.3390/children10040647

**Published:** 2023-03-30

**Authors:** Maísa Malta, Rowim AlMutiri, Christine Saint Martin, Myriam Srour

**Affiliations:** 1Research Institute of the McGill University Health Center, Montreal, QC H4A 3J1, Canada; 2Division of Child Neurology, Department of Neurology and Neurosurgery, Federal University of São Paulo, São Paulo 04024-002, Brazil; 3Division of Pediatric Neurology, Department of Pediatrics, McGill University, Montreal, QC H4A 3J1, Canada; 4National Neuroscience Institute, King Fahad Medical City, Riyadh 12231, Saudi Arabia; 5Department of Radiology, McGill University, Montreal, QC H4A 3J1, Canada

**Keywords:** holoprosencephaly, nervous system malformation, craniofacial abnormalities, neurodevelopmental disorders, SHH, sonic hedgehog, genetic testing

## Abstract

Holoprosencephaly (HPE) is the most common malformation of the prosencephalon in humans. It is characterized by a continuum of structural brain anomalies resulting from the failure of midline cleavage of the prosencephalon. The three classic subtypes of HPE are alobar, semilobar and lobar, although a few additional categories have been added to this original classification. The severity of the clinical phenotype is broad and usually mirrors the radiologic and associated facial features. The etiology of HPE includes both environmental and genetic factors. Disruption of sonic hedgehog (SHH) signaling is the main pathophysiologic mechanism underlying HPE. Aneuploidies, chromosomal copy number variants and monogenic disorders are identified in a large proportion of HPE patients. Despite the high postnatal mortality and the invariable presence of developmental delay, recent advances in diagnostic methods and improvements in patient management over the years have helped to increase survival rates. In this review, we provide an overview of the current knowledge related to HPE, and discuss the classification, clinical features, genetic and environmental etiologies and management.

## 1. Introduction

Holoprosencephaly (HPE) is characterized by a continuum of structural anomalies of the brain resulting from the failure of differentiation and midline cleavage of the prosencephalon (i.e., forebrain) during the third to fourth weeks of gestation [1,2,3,4,5,6]. HPE is the most common malformation of the prosencephalon in humans. It affects 1:250 conceptuses; however, given the significant number of fetal deaths, its prevalence in live births is lower, ranging from 1:8000 to 1:16,000. This rate appears to be consistent across different populations [7,8].

The degrees of severity of HPE are defined by the extent of the brain malformation, and are classically divided into alobar, semi-lobar, and lobar forms. A spectrum of midline craniofacial defects are associated with HPE [2]. The severity of the clinical phenotype usually mirrors the radiologic and associated facial features [2]. Important advances in the understanding of the etiologic factors, especially the genetic basis, have been achieved in recent decades [9].

In this review, we will present a brief overview of prosencephalic development, and discuss the classification, clinical features, etiology and management related to HPE.

## 2. Overview of Prosencephalic Development

Prosencephalic development consists of three sequential phases that usually overlap, namely, the formation, cleavage, and midline development [10]. Soon after the closure of the anterior neuropore, between days 22 and 24 of gestation, the embryonic prosencephalic vesicle is established at the rostral end of the neural tube. As the prosencephalon undergoes expansion, the cells located at the midline undergo high levels of death and reduced proliferation leading to the splitting of the expanding telencephalon (which will give rise to the cerebral hemispheres) and the diencephalon (which will give rise to the thalamus) [1].

Normal face and brain development require an adequate balance of dorsalizing and ventralizing factors in the developing prosencephalon. Key signaling molecules for the patterning of the prosencephalic midline include bone morphogenic proteins (BMPs), wingless-integrated proteins (WNTs), fibroblast growth factors (FGF) and sonic hedgehog proteins (SHH), which are secreted in the dorsal, rostral and ventral midlines areas**,** respectively. SHH is a secreted protein that has a key role in the maintenance of the notochord and the patterning and induction of the ventral forebrain [11]. Hedgehog (Hh) signal transduction requires the integrity of the cilia, microtubule-based organelles that project from the cell’s surface, helping the traffic of the Hh pathway proteins [12]. Disruption of midline patterning, as seen in HPE, results in the failure of prosencephalon cleavage into distinct right and left hemispheres. Deep brain structures, and the olfactory and optic bulbs and tracts, can also be affected [3,4,5,6].

The development of the face occurs simultaneously with the formation of the forebrain and relies on signaling from the ventral midline mediated by SHH. The cranial neural crest cells, which are derived from the ectoderm situated at the dorsal border between the neural tube and surface ectoderm of the embryo, migrate towards the pharyngeal arches and the frontonasal process to form the tissues of the skull, the upper cervical tract, and the face [13]. Disruption of this process results in craniofacial defects, as seen in patients with HPE [2]. In cases of severe holoprosencephalies, the premaxillary segments of the face remain unformed, causing midline facial defects, such as arrhinia, midline facial clefts, hypotelorism, and cyclopia [11,14].

## 3. Radiologic Classification of HPE

The widely accepted classification system for HPE was proposed by DeMyer, who divided holoprosencephaly into three subcategories: alobar, when there is a complete lack of separation of the cerebral hemispheres and a large monoventricle (Figure 1a–d); semi-lobar, with only the anterior lobes failing to separate but the parieto-occipital regions divided by the interhemispheric fissure and falx cerebri (Figure 1e–g); and lobar, when only the most rostral-inferior parts of the frontal lobes are fused [15,16].

Since then, a few more categories have been added to the original classification. The middle interhemispheric (MIH) variant is characterized by the separation of the anterior and occipital regions of the brain’s hemispheres, while the posterior frontal and parietal lobes remain fused [17] (Figure 1h–j). Milder, minimal and microforms of HPE are also described and are, respectively, associated with septo-optic dysplasia, nonseparation of the preoptic area, and only facial features (hypotelorism and a single maxillary central incisor) with normal brain development [10,15,18]. The radiologic features of holoprosencephalies are summarized in Table 1.

It is fundamental to keep in mind that holoprosencephalies represent a continuum of forebrain malformations and no clear distinctions among these different categories should be expected [23,24]. In some cases of lobar HPE, the fusion between the frontal lobes may be minimal, making it particularly difficult to differentiate between a mild lobar HPE and other midline malformations (such as septo-optic dysplasia or isolated fusion of the fornices). A useful radiologic sign on doppler ultrasound supporting the presence of a mild HPE is the anterior displacement of the anterior cerebral artery underneath the frontal bone, giving an appearance of a serpent under the calvarium, also known as the “snake under the skull” sign [2,25].

## 4. Clinical Features

### 4.1. Craniofacial Defects and Developmental Outcomes

The spectrum of craniofacial defects in patients with holoprosencephaly can range from mild to very severe, and are an indicator of the severity of the intracranial lesion [2]. Overall, developmental delay is present in virtually all individuals within the spectrum of holoprosencephalies; however, the degree of severity is variable and correlates with the extent of brain malformations and craniofacial defects [26].

#### 4.1.1. Alobar HPE

The most severe facial malformations are associated with Alobar HPE, which affects the development of the eyes, nose, upper lip, and palate. In this variant, the eyes and optic nerves, as well as the olfactory bulbs and tracts, may be absent, fused or separated [19]. The ophthalmologic features include cyclopia (single central eye), synophthalmia (two fused eyes in the midline), and hypotelorism. Vision is usually impaired in these severe cases associated with ocular malformations. Optic nerve hypoplasia and iris or uveoretinal colobomas are common [20]. Defects in the nasal region may present as a complete absence of the nose with a proboscis (also known as ethmocephaly), or as a small nose with a single nostril (referred to as cebocephaly). Those with nasal defects may also have oral anomalies, such as a midline cleft lip and/or palate [14], which can additionally contribute to feeding difficulties.

Individuals with alobar HPE who survive the neonatal period uniformly have profound developmental impairment. Motor deficits are severe and the affected individuals usually have limb spasticity and axial hypotonia [3]. Affected individuals do not achieve the ability to sit independently. In terms of fine motor skills, they typically can only reach and bat at objects. They have minimal hand function. The degree of non-separation of the deep gray nuclei, especially the caudate and lentiform, correlates with the common emergence of extrapyramidal features, such as dystonia and choreoathetosis [3]. Furthermore, individuals of this group are nonverbal with profound cognitive impairment. These patients are fully dependent on all activities of daily living [6,27].

#### 4.1.2. Semilobar HPE

For patients with semilobar HPE, the facial anomalies and developmental outcomes can be extremely variable. Some patients will show more severe facial abnormalities, such as anophthalmia/microphthalmia, absent nasal septum, and lip/palate clefts, while others may show a relatively normal facial appearance [16].

Similar to the craniofacial defects, the motor outcomes in children with semilobar HPE are also variable and can range from more severe deficits, where assistance is required for sitting and the individuals are usually wheelchair-dependent, to the achievement of independent ambulation in some other cases [3]. Dystonia and spasticity can be variable. The majority of children with semilobar HPE, however, have severe motor deficits [6]. Most individuals with semilobar HPE are non-verbal, though rare individuals can say words [3]. They usually have a profound intellectual disability [27].

#### 4.1.3. Lobar HPE and MIH Variant

In mild cases of HPE, the subtle facial anomalies can be easily overlooked. Among the facial features seen in patients with lobar HPE are a bilateral cleft lip with a median process, closely spaced eyes, and a depressed nasal bridge [6,17,21]. Even so, a relatively normal facial appearance is the most common finding in this group of patients and, therefore, they are quite often diagnosed later, upon the emergence of developmental delay, epilepsy or movement disorders [28,29].

In addition, patients with the MIH variant usually do not present any associated craniofacial anomalies [22].

Patients who have the lobar and mild forms of HPE typically have a better neurodevelopmental outcome than the alobar/semilobar subtypes. These children often have less significant dystonia and spasticity [30], and can either walk independently or with assistance [3].

### 4.2. Survival

HPE is associated with high postnatal mortality. Overall, the estimated mortality rate for all subtypes of HPE is 33% in the first 24 h after birth, and 58% in the first month [31,32]. The reported survival rate after 1 year of life is around 29% [31].

High mortality rates were also reported specifically in patients with the lobar variant [33]. In another study, including a cohort of children with alobar HPE and mild or no facial malformations, the mortality rate was about 50% by the age of 4–5 months [34]. In addition to the subtype, survival rates have also been correlated with genetic findings, as non-syndromic, euploid HPE patients showed overall better outcomes [16].

Despite the high mortality seen in patients with alobar HPE, patients with less severe variants, including the semilobar and lobar variants, seem to have a better overall survival rate. A cohort of adolescents and adults with HPE showed that 50% had the semilobar variant. Interestingly, none of the patients included displayed the typical facial features [27], enhancing the idea that the lesser extent of craniofacial defects also correlates with higher life expectancies. A significant proportion of patients on the milder HPE spectrum survive beyond 12 months of age [6].

The improvement of the survival rates of patients with HPE in recent years has been attributed to the advances in diagnostic methods and in patient management over the years [26,30,35].

### 4.3. Other Common Clinical Features

Hydrocephalus is a frequent complication in children with HPE (seen in 16 to 40% of the cohort [27]) and it is most common in individuals with alobar HPE, requiring shunt placement [27]. Almost all individuals without hydrocephalus will have microcephaly.

Aside from the midline cerebral anomalies described above, delayed white matter maturation is often described on the brain imaging of individuals with HPE, especially in those with the alobar subtype [3]. Cortical dysplasia and subcortical heterotopias are also frequently present, most commonly in the MIH variant subtype [16].

Seizures can occur in approximately 50% of HPE patients and, apparently, there is no correlation between the severity of HPE and its occurrence [30]. No particular type of seizure is considered characteristic of this condition and the frequency of seizures may vary during the course of the evolution. Seizures are usually responsive to anticonvulsant treatment but some cases can be medically refractory, especially when they are associated with cortical dysplasia [30].

Other common clinical manifestations observed in children with HPE include hypothalamic disorders, leading to the instability of temperature and heart rate, and brainstem dysfunction, leading to swallowing and respiratory difficulties [6,26,36]. Pituitary dysfunction with central diabetes insipidus is the most common finding in patients with non-chromosomal, non-syndromic HPE. Short stature and a failure to thrive are also frequent, especially in more severely affected children [28,36]. In a cohort of 117 patients with HPE hypoadrenocorticism was present in 7% and GH deficiency in 5% of patients [37].

Feeding difficulties are also a major problem due to a combination of factors that include brainstem dysfunction, axial hypotonia, lethargy, seizures, medication side effects, and lack of interest. Gastrointestinal tract anomalies, such as poor gastric and colonic motility and gastroesophageal reflux, are often present [26]. The majority of children with alobar and semilobar HPE are nourished through a gastrostomy tube [27].

## 5. Etiology

HPE is caused by both genetic and non-genetic risk factors (e.g., teratogens), and the interactions between these are likely responsible for the complex variation in outcomes of the HPE phenotypes [5,38,39,40]. A summary of the factors involved in the etiology of HPE is provided in Table 2 and Table 3.

### 5.1. Genetic Causes

Genetic causes of HPE include chromosomal abnormalities and single gene disorders. Individuals with syndromic forms of HPE (i.e., associated with other congenital anomalies and organs or systems involvement) are typically associated with chromosomal anomalies or single gene disorders; whereas, non-syndromic or isolated forms of HPE are more likely to have a monogenic cause [2,6].

#### 5.1.1. Chromosomal Anomalies

Approximately 25–50% of individuals with HPE have a chromosomal abnormality (aneuploidy or structural anomaly) [52]. Trisomy 13 (Patau syndrome) is by far the most common cause of HPE, being responsible for 75% of HPE-related chromosomal abnormalities [6]. Trisomy 18 (Edwards syndrome) and various other aneuploidies are also found in association with HPE [53].

Pathogenic structural chromosome abnormalities, i.e., copy number variants (CNVs), are found in 10–14% of individuals with HPE [6,52,54]. Though they have been reported in virtually all chromosomes, causal structural chromosomal anomalies frequently involve regions harboring known HPE genes. For example, CNVs involving chromosome 18p11.3, which include *TGIF1*, a HPE gene, have been reported in many patients [54]. Similarly, recurrent CNVs in 2p21, which includes *SIX3*, have been published [54].

#### 5.1.2. Monogenic Causes

A monogenic etiology is identified in approximately 18–25% of individuals with HPE [52,55]. These include genes associated with both non-syndromic and syndromic forms of HPE. All HPE genes identified to date are involved directly or indirectly in the regulation of the SHH pathway [56].

Non-syndromic monogenic causes

To date, there are at least 17 genes known to be associated with non-syndromic HPE (see Table 2). Of these, *SHH*, *ZIC2*, *GLI2*, *SIX3*, *FGF8* and *FGFR1* are considered major HPE genes, as they are each responsible for more than 2% of HPE cases [35].

*SHH* was the first HPE gene to be discovered [57]. Heterozygous pathogenic variants in *SHH* are the most common monogenic cause of non-syndromic HPE, responsible for 5.4% to 5.9% of cases [38,52,58,59,60]. *SIX3* (Six Homeobox 3), which acts as a direct regulator of SHH expression in the anterior diencephalon [61], is responsible for around 3% of cases [52]. Pathogenic variants in the transcription factors *ZIC2* and *GLI2* are found in 4.8–5.2% [11,36,62] and 3.2% [63,64,65] of HPE cohorts.

A few genotype–phenotype correlations have been noted. The most severe types of HPE (alobar and semi lobar) are more frequently associated with the *ZIC2* and *SIX3* variants [52]. Most patients identified with *ZIC2* variants do not display typical HPE facial anomalies, showing more subtle features that include bitemporal narrowing, up slanting palpebral fissures, a flat nasal bridge, a short nose with anteverted nares, a broad and deep philtrum, and a subjective appearance of large ears [36]. *SHH* and *GLI2* tend to be more frequently associated with HPE microforms, with *GLI2* frequently found in patients with additional pituitary anomalies [52].

Pathogenic variants in non-syndromic HPE genes are primarily inherited in an autosomal dominant fashion, with incomplete penetrance and variable expressivity [6]. De novo variants are often found in sporadic HPE cases. In familial forms, there is usually wide variability in the phenotype of affected family members. Pathogenic variants can be inherited from a parent who is asymptomatic or has a very subtle phenotype that may not have been previously recognized. For example, only approximately one-third of individuals with *SHH* pathogenic variants have HPE, while the rest may display microforms or have no signs at all [52,66]. This highlights the challenge and complexity of genetic counseling in families with HPE. The observation of wide intra-familial variability has led to the hypothesis of an oligogenic mode of inheritance in non-syndromic HPE genes, where the penetrance and expressivity of a variant is modulated by variants in additional genes associated with HPE [52]. Furthermore, additional environmental modifiers are thought to impact expressivity and penetrance [3,39,52].

Syndromic monogenic causes

Syndromic monogenetic HPE can be inherited in either a dominant or recessive manner. The following are some syndromic disorders commonly associated with HPE.


*Smith–Lemli–Opitz syndrome (SLOS)*


A classic example of a monogenic disorder related to syndromic HPE is Smith–Lemli–Opitz syndrome (SLOS), associated with biallelic pathogenic variants *DHCR7* [67,68,69,70,71]. SLOS is caused by a deficiency of the enzyme 7-dehyrocholestol (DHC) reductase, resulting in a block in the last step of cholesterol synthesis [55]. Cholesterol modification of the *SHH* ligand is required for its full signal transduction activity [72]; therefore, low cholesterol has been proposed to affect the SHH signaling pathway. Clinically, patients with SLOS typically present characteristic facial features, growth delay, microcephaly, polydactyly, syndactyly of the second and third toes, cleft palate, underdeveloped external male genitalia and intellectual disability [71].


*Steinfield syndrome*


Steinfeld syndrome is an autosomal dominant disorder characterized by HPE associated with limb defects, although other anomalies can also be present [73]. Its causal gene, *CDON*, encodes a hedgehog receptor.


*FGFR1-related syndromes*


FGFR1 encodes fibroblast growth factor receptor 1. The fibroblast growth factor family is important for gonadotropin releasing hormone neuronal development. Two syndromes related to *FGFR1* mutations are Kallmann and Hartsfield.

Clinical manifestations of *FGFR1* alterations are very heterogeneous and can result in two autosomal dominant syndromes that can be associated with HPE: Kallmann syndrome, which is characterized by hypogonadotropic hypogonadism and anosmia; and Hartsfield syndrome, which is associated with HPE, ectrodactyly, cleft lip and palate [74,75,76,77].


*Stromme syndrome*


Stromme syndrome is a rare cause of ciliopathy with a characteristic triad that includes intestinal atresia, microcephaly and variable ocular abnormalities, which may be seen in association with HPE [78]. Its causal gene, *CENPF*, encodes a centromeric protein and has an important role in ciliogenesis, which may impact SHH signaling [79].

### 5.2. Non-Genetic Factors

Maternal diabetes (both gestational and pre-gestational) is the most well-established risk factor for HPE. Pre-gestational diabetes requiring insulin offers more than a 10-fold increased risk [41,80]. A prevalence of 1% to 2% of HPE in infants of diabetic mothers has been reported [42].

Exposure to alcohol is another known risk factor. In a recent study, the number of alcoholic drinks consumed per week during pregnancy was associated with increased HPE risk [43].

Animal model findings are consistent with these observations, as gestational diabetes and prenatal alcohol exposure in mice were clearly associated with HPE [46,47,48,49]. Oxygen free radicals, apoptosis, and inhibition of neural crest cell migration and differentiation are suggested teratogenic mechanisms involved in both maternal diabetes and alcohol exposure [44].

Other teratogenic agents that directly or indirectly disturb the SHH pathway have been shown to be associated with the occurrence of HPE in humans and animal models. These include retinoic acid, food-borne mycotoxins (such as ochratoxins), cyclopamine (an SHH signaling inhibitor), and drugs that interfere with cholesterol biosynthesis [45,50].

Exposure to heavy metals and radiation were also suggested to have deleterious effects on brain development, although no causality has been clearly established with HPE. For example, the occupational exposure to X-rays was correlated with an increased severity in HPE cases [43], even though the maternal exposure to radiation was not associated with a higher prevalence [43,51,81]. Similarly, an African case series suggested that teratogenicity related to mining-related pollution could be responsible for the emergence of three cases of HPE [82], although the potential connection between metal exposure and HPE occurrence has not been properly investigated, yet.

Consistent maternal folic acid was reported as a protective factor against HPE, although previous studies have shown mixed results [2,51]. In one study, folic acid ingestion decreased the risk of HPE by as much as 73% when taken during the first month of pregnancy [43].

## 6. Investigations and Management

### 6.1. Brain Imaging

The more severe types of holoprosencephaly (alobar, semilobar) can already be detected using prenatal ultrasound in the first trimester [2,83]. Imaging findings may reveal the complete or partial absence of the interhemispheric fissure, distorted appearance of the choroid plexuses in the axial transventricular plane, and fused thalami, as well as midline facial anomalies [16]. However, milder degrees of HPE, including lobar and some forms of semilobar HPE, cannot be reliably detected using fetal ultrasound. When ultrasound studies have indicated the presence of a potential cerebral anomaly, fetal MRI is often used as a second-line investigation to confirm the findings and evaluate the central nervous system’s structure [84] (Figure 2a-c).

HPE is most frequently diagnosed during the newborn period when abnormal facial findings and/or a neurologic presentation lead to further imaging and evaluation [2,30]. The imaging modality of choice is the MRI, which allows a definition of the type of HPE and the identification of any associated brain anomalies [18,83,85] (Figure 2d–f). It is crucial that the imaging be reviewed by a radiologist or another clinician who is knowledgeable about the clinical subtypes of HPE, as subtle midline anomalies may be overlooked. Similarly, neuroimaging abnormalities not related to HPE, such as callosal dysgenesis, arrhinencephaly, and pituitary dysgenesis, may be mistaken for findings of HPE [3,86].

### 6.2. Genetic Testing

Genetic testing should be performed in all individuals with HPE and should start with a chromosomal analysis. If there is a clinical suspicion of an aneuploidy, such as trisomy 13 or 18, the investigation should begin with a karyotype; otherwise, a chromosomal microarray (CMA) should be performed first [35,87].

When a specific syndromic cause is suspected, a sequence analysis of the gene of interest should be performed. However, if an etiology is not readily apparent in an individual with syndromic HPE, a comprehensive developmental disorders gene panel or exome sequencing should be performed after an unrevealing chromosomal analysis.

For individuals with non-syndromic HPE, testing of the known HPE genes with a multigene panel is a reasonable diagnostic approach [6]. Alternatively, exome sequencing can also be considered. In a recent study, exome sequencing in a cohort of HPE patients (both syndromic and non-syndromic) who had previously had unrevealing genetic investigations reported an overall yield of 22% [88].

### 6.3. Considerations Related to Prenatal Diagnosis and Genetic Counseling

In the event of a prenatal diagnosis of HPE on ultrasound or fetal MRI, genetic testing can also be performed during pregnancy for the identification of chromosomal abnormalities or monogenic disorders associated with HPE. Fetal karyotype, CMA analyses or even multigene panels are possible through the analysis of amniotic fluid [6].

By the time a prenatal diagnosis of HPE is typically made, the pregnancy has usually reached 18 to 22 weeks [89], facing the parents with very challenging decisions within a narrow timeframe, such as the consideration of pregnancy termination and/or the establishment of postnatal goals of care for the unborn child [90].

A study conducted with mothers who received a prenatal diagnosis of alobar HPE revealed that parents’ needs for information, emotional support, and assistance are often not being adequately addressed by the healthcare providers [90]. Usually, parents are given a very negative prognosis and they often expect that their children will not survive beyond a few days or weeks after birth [91]. However, the prognosis of these patients is often uncertain, especially for those with less severe forms.

Recommended practices for managing these cases include the avoidance of terms such as “not viable”, “incompatible with life”, and “vegetable”; the provision of balanced information on the possible outcomes (including the chance of survival beyond the perinatal period) based on both the literature and other parental experiences; and referral to a specialist [90].

Meeting with a genetic counselor or a geneticist is also recommended to assist in the estimation of recurrence risk in future pregnancies [90]. Overall, it is estimated that the recurrence risk can be up to 50% in the case of a parent carrying a mutation with incomplete penetrance and 1% in the case of an aneuploidy such as trisomy 13 [30]. The counseling of families with pathogenic variants in non-syndromic HPE genes involves a review of the concepts related to incomplete penetrance and variable expressivity.

In cases where a clinical diagnosis is established but the genetic cause is not known, it is not possible to predict the recurrence risk for future pregnancies [89,92]. That imposes a lot of stress, and a sense of uncertainty, on the parents in the face of familial planning decisions.

### 6.4. Management

It is highly recommended that the care of patients with HPE be based on a multidisciplinary approach, with the involvement of gastroenterologists, neurologists, neurosurgeons, and plastic surgeons, among other health care professionals.

A detailed ophthalmologic evaluation and hearing assessment should be performed in all patients. Feeding and neurodevelopment should be closely followed. Complications related to oromotor dysfunction, aspiration, endocrinologic dysfunction, hydrocephalus and seizures are frequent and should be monitored [26].

Pharmacological treatment of gastroesophageal reflux may be necessary, and, in severe cases, Nissen fundoplication or transpyloric feed can be recommended [30]. When indicated, gastrostomy tube placement might help prevent respiratory complications and reduce hospitalization rates.

Evaluation by a pediatric endocrinologist should be performed on a regular basis, given the common occurrence of endocrinologic dysfunction [2].

Movement disorders and prominent spasticity with possible progression to orthopedic deformities are often present and may become more prominent during the follow-up period. The management of these conditions might resemble that of patients with cerebral palsy, and the inclusion of physical and occupational therapy represent good preventive measures [30]. Cases with refractory pain or functional limitations might benefit from intramuscular botulin toxin injections, oral antispasmodic agents, or even orthopedic surgery for those with complications, such as contractures, hip dislocation and scoliosis [30].

Challenges and needs of the patients evolve over their lifespan. Individuals with HPE who are higher functioning may experience neuropsychologic problems, including executive dysfunction, attention deficit hyperactivity disorder, and learning disabilities [30]. Psychologic/psychiatric intervention is frequently the first line of approach recommended; however, upon the eventual emergence of complications (i.e., anxiety and depression), pharmacologic treatment may be necessary.

## 7. Conclusions

In summary, holoprosencephaly is a continuum of structural brain anomalies resulting from the failure of midline cleavage of the prosencephalon. The spectrum and severity of the structural malformations as well as the associated clinical features is broad. In general, the craniofacial features are a good indicator of the severity of the intracranial lesion and clinical outcome. Remarkable progress has been made in understanding the etiology of HPE, with disruption of SHH signaling either from genetic or environmental (teratogens) factors being the main mechanism underlying pathophysiology. A genetic etiology, either from a chromosomal aberration or monogenic cause, can be identified in the majority of individuals. For families with autosomal dominant non-syndromic HPE, genetic counseling remains challenging given the large variability in penetrance and expressivity. A multidisciplinary approach remains a cornerstone in the clinical management of patients.

## Figures and Tables

**Figure 1 children-10-00647-f001:**
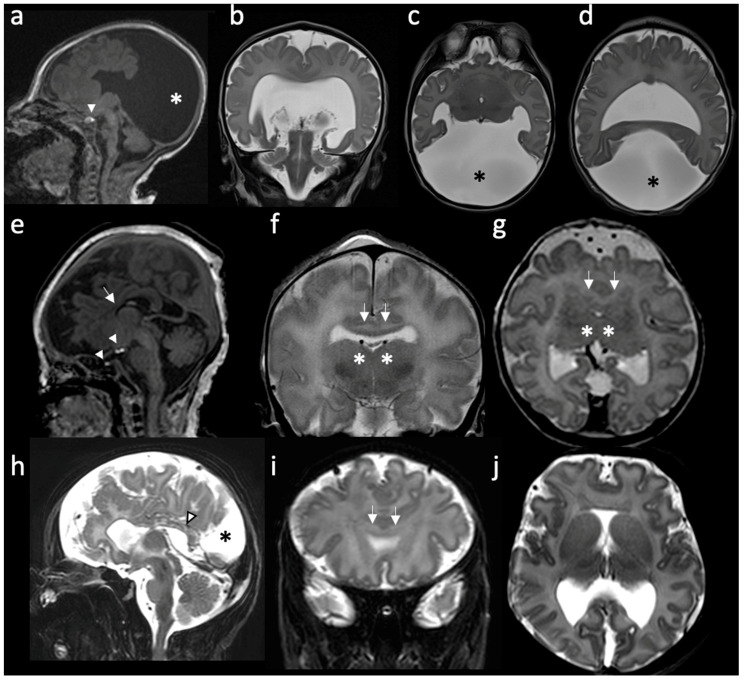
Radiologic features of holoprosencephalies: (**a**–**d**) Radiologic features of alobar holoprosencephaly, 2 days old. (**a**) Sagittal T1-weighted image shows hypodevelopment of the parietal lobes with absence of identifiable occipital lobes and corpus callosum. The bright T1 spot of the neurohypophysis is preserved and well placed (white arrowhead). (**b**) Coronal T2-weighted image shows the frontal lobes fused across the midline and a large supratentorial monoventricle. (**c**,**d**) Axial T2-weighted images show fused thalami and the monoventricle communicating with a large dorsal cyst (asterisk), noting the absence of the septum pellucidum and rudimentary formation of the temporal horns. (**e**–**g**) Radiologic features of semilobar holoprosencephaly, 3 days old. (**e**) Sagittal T1-weighted image shows hypoplastic frontal lobes, absence of anterior corpus callosum (white arrow) and abnormal finding of partial ectopic neural hypophysis associated with residual hyperintense signal seen within the sella (white arrowheads). (**f**) Coronal T2-weighted image shows fusion of the frontal lobes (white arrows) and partial fusion of thalami (asterisk). (**g**) Axial T2-weighted image shows that the division of the ventricles is only seen posteriorly. Septum pellucidum is absent. (**h**–**j**) Radiologic features of middle interhemispheric variant holoprosencephaly, 15 days old. (**h**) Sagittal T2-weighted image shows absent body of the corpus callosum but with the presence of the splenium (white arrowhead). Dorsal interhemispheric cyst is present (black asterisk). (**i**) Coronal T2-weighted image shows fusion of the frontal lobes across the midline (white arrows). Note that the degree of fusion is less extensive than the one seen in the semilobar HPE. (**j**) Axial T2-weighted image shows fused frontal lobes, absent septum pellucidi and the interhemispheric cyst in the occipital region.

**Figure 2 children-10-00647-f002:**
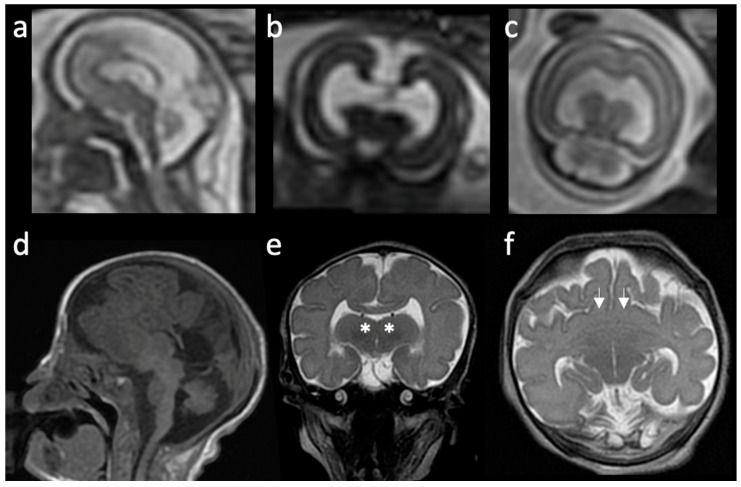
Pre- and postnatal imaging findings of a patient with holoprosencephaly: (**a**–**c**) Prenatal MRI of a patient with semilobar holoprosencephaly. (**a**) Sagittal T2-weighted image shows a mass of parenchymal mantle anteriorly, and partial development of occipital lobes. (**b**) Coronal T2-weighted image shows a large monoventricle and thalami lying very close to one another. (**c**) Axial T2-weighted image shows a supratentorial monoventricle but with two distinct occipital horns. (**d**–**f**) Postnatal MRI of the same patient, 3 days old. (**d**) Sagittal T1-weighted image shows hypoplastic occipital lobes, partially absent anterior corpus callosum. (**e**) Coronal T2-weighted image shows bilateral fusion of the bilateral thalami (white asterisk) with severe hypodevelopment of the basal ganglia. (**f**) Axial T2-weighted image shows fusion of the frontal lobes (white arrows) and abnormal orientation of the rudimentary sylvian fissures with diffused decreased degree of sulcation.

**Table 1 children-10-00647-t001:** Radiologic classification of holoprosencephaly.

Type	Main Features	References
Alobar	Absent separation of the cerebral hemispheres; Single “monoventricle”; Agenesis of the corpus callosum, absent third ventricle; Fusion of thalami and basal ganglia; Dorsal cyst is frequent; Significant midline facial defects.	[6,14,16,19,20]
Semilobar	Anterior lobes fail to separate; Interhemispheric fissure detected only posteriorly; Small, partially-formed third ventricle is often noted; Dorsal cyst may also be present; Midline craniofacial defects may be present or only subtle facial abnormalities.	[6,16]
Lobar	Only the most rostral-inferior parts of the frontal lobes are fused; Septum pellucidum is usually absent; Posterior half of the corpus callosum is formed; Varying degrees of basal ganglia and thalamic fusion; Midline craniofacial defects often absent or mild.	[6,16,21]
Middle interhemispheric variant (syntelencephaly)	Failure of separation of the posterior frontal and parietal lobes; Variable lack of cleavage of the basal ganglia and thalami; Absence of the body but presence of the genu and splenium of the corpus callosum.	[6,16,17,22]
Septopreoptic (minimal form)	Midline fusion restricted to the septal region or preoptic region of the telencephalon.	[16]
Microform	Only HPE-related subtle craniofacial anomalies; No structural brain defects on imaging.	[16]

**Table 2 children-10-00647-t002:** Summary of common genetic causes of holoprosencephaly.

Genetic Causes			
Syndromic			Non-Syndromic
Chromosomal		Monogenic	
Aneuploidies	Structural Abnormalities		
Trisomy 13 (Patau syndrome) Trisomy 18 (Edwards syndrome)	del or dup(13q) del(18p) del(7)(q36) dup(3)(p24-pter) del(2)(p21) del(21)(q22.3)	*CDON*-Steinfeld syndrome	*SHH* (AD, MIM#142945)
		(AD, MIM#184705)	*ZIC2* (AD, MIM#609637)
		*FGFR1*-Kallmann syndrome 2	*SIX3* (AD, MIM#157170)
		(AD, MIM# #147950)	*TGIF1* (AD, MIM#142946)
		*FGFR1*-Hartsfield syndrome	*GLI2* (AD, MIM#610829)
		(AD, MIM#615465)	*FGF8* (AD, MIM#612702)
		*CENPF*-Stromme syndrome	*FGFR1* (AD, MIM#147950)
		(AR, MIM#243605)	*DISP1* (AD, MIM#612530)
		*DHCR7*-Smith-Lemli-Optiz syndrome	*DLL1* (AD, MIM#618709)
		(AR, MIM#270400)	*CDON* (AR, MIM#614226)

AD: autosomal dominant; AR: autosomal recessive.

**Table 3 children-10-00647-t003:** Summary of common non-genetic risk factors related to holoprosencephaly.

Non-Genetic Risk Factors	References
Maternal diabetes	[41,42,43,44,45]
Alcohol exposure	[44,45,46,47,48,49]
Retinoic acid	[50]
Food-borne mycotoxins	[50]
Cyclopamine	[50]
Drugs that interfere with cholesterol biosynthesis	[51]

## Data Availability

Not applicable.

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
