# Peer review of "Holoprosencephaly: Review of Embryology, Clinical Phenotypes, Etiology and Management"

_children, 2023, doi:10.3390/children10040647_

Round 1

Reviewer 1 Report

A review about the complex topic of holoprosencephaly (HPE) is very helpful, as it provides a basis for understanding of the different conditions causing HPE. An update would be especially useful. The causes of HPE are probably different from a few decades ago.   For example, trisomy 13 (mentioned as causing 75% of cases in section 5.1.1) is probably no longer such a major cause of HPE as more aetiological factors causing HPE have been discovered. 

There are minor spelling errors, for example in the first line of the second paragraph in the Introduction - 'The degrees of severity of HPE are fined by the extent of the brain malformation' should be 'the degrees of severity of HPE are defined by the extent....'.

The authors could provide a reference for Table 1. 

In Section 4, more information about prognosis and survival in different categories of HPE would be useful, for example, some types of HPE are usually fatal while others have much better outcome. Brief summaries of the more common conditions associated HPE would be helpful. 

In section 5, some of the monogenic causes are related to the SHH signalling pathway and these could be listed, such as GLI2. The role of cilia in SHH signalling could also be outlined.

Author Response

We would like to thank your valuable and constructive comments. Please find below a point-by-point response to the comments.

  1. A review about the complex topic of holoprosencephaly (HPE) is very helpful, as it provides a basis for understanding of the different conditions causing HPE. An update would be especially useful. The causes of HPE are probably different from a few decades ago.   For example, trisomy 13 (mentioned as causing 75% of cases in section 5.1.1) is probably no longer such a major cause of HPE as more aetiological factors causing HPE have been discovered. 
  • Thank you for your comments. Actually, the Trisomy 13 remains as a very frequent cause of HPE and the figures we have provided above are correct. The metaanalysis by Paul Kruszka et al in 2018 found that Trisomy 13 was the most common cause of HPE, observed in 40%-60% of HPE of all causes and about 75% of HPE caused by chromosome abnormalities [PMID: 29770994]. These figures are also used in the holoprosencephaly overview published in genereviews, last updated in 2020 [https://www.ncbi.nlm.nih.gov/books/NBK1530/]. 
  1. There are minor spelling errors, for example in the first line of the second paragraph in the Introduction - 'The degrees of severity of HPE are fined by the extent of the brain malformation' should be 'the degrees of severity of HPE are defined by the extent....'.
  • Thank you for pointing that out. That issue has been corrected.
  1. The authors could provide a reference for Table 1. 

As requested, we have provided references for Table 1.

  1. In Section 4, more information about prognosis and survival in different categories of HPE would be useful, for example, some types of HPE are usually fatal while others have much better outcome. Brief summaries of the more common conditions associated HPE would be helpful. 
  • Thank you for your suggestion. Accordingly, we made a few changes in the section about survival to better address the different rates in the HPE subtypes: (p.5; l.190-208)

(…) HPE is associated with high postnatal mortality. Overall, the estimated mortality rate for all subtypes of HPE is 33% in the first 24 hours after birth, and 58% in the first month [31,32]. The reported survival rate after 1 year of life is around 29% [31].

High mortality rates were also reported specifically in patients with the lobar variant [33]. In another study with children with alobar HPE and mild or no facial malformations, the mortality rate was about 50% by the age of 4– 5 months [34]. Besides the subtype, survival rates have also been correlated with genetic findings, as non-syndromic, euploid HPE patients showed overall better outcomes [16].

Despite the high mortality seen in patients with alobar HPE, patients with less severe variants, including semilobar and lobar, seem to have a better overall survival rate. A cohort of adolescents and adults with HPE showed that 50% had the semilobar variant. Interestingly, none of the patients included displayed the typical facial features [25], enhancing the idea that the lesser extent of craniofacial defects also correlates with higher life expectancies. A significant proportion of patients on the milder HPE spectrum survives beyond 12 months of age [6].

The improvement of the survival rates of patients with HPE in recent years has been attributed to advances in diagnostic methods and in patient management over the years [22,30,35] (…).

  • Common syndromes associated with HPE were included in the Etiology ® monogenic causes ® Chromosomal anomalies and Syndromic monogenic sections.
  1. In section 5, some of the monogenic causes are related to the SHHsignalling pathway and these could be listed, such as GLI2. The role of cilia in SHH signalling could also be outlined.
  • We had already listed GLI2 both in the non-syndromic monogenic causes section and Table 2, as shown (p.9; l. 283-291):

(…) To date, there are at least 17 genes known to be associated with non-syndromic HPE (see table 2). Of these, SHH, ZIC2, GLI2, SIX3, FGF8 and FGFR1 are considered major HPE genes, as they are each responsible for more than 2% of HPE cases [35].

SHH was the first HPE gene to be discovered [41]. Heterozygous pathogenic variants in SHH are the most common monogenic cause of non-syndromic HPE, responsible for 5.4% to 5.9% of cases [33,36,42–44]. SIX3 (Six Homeobox 3), which acts as a direct regulator of SHH expression in the anterior diencephalon [45], is responsible for around 3% of cases [36]. Pathogenic variants in the transcription factors ZIC2 and GLI2 are found in 4.8%-5.2% [11,27,46] and 3.2% [47–49] of HPE cohorts (…).

  • As suggested, we expanded the discussion on the role of cilia in the SHH signaling pathway in the overview of prosencephalic development section (2; l. 55-66):

(…) Normal face and brain development require an adequate balance of dorsalizing and ventralizing factors in the developing prosencephalon. Key signaling molecules for the patterning of the prosencephalic midline include bone morphogenic proteins (BMPs), wingless-integrated proteins (WNTs), fibroblast growth factors (FGF) and sonic hedgehog (SHH), which are secreted in the dorsal, rostral and ventral midlines, respectively. SHH is a secreted protein that has a key role in the maintenance of the notochord and the patterning and induction of the ventral forebrain [11]. The Hedgehog (Hh) signal transduction requires the integrity of the cilia, a microtubule-based organelle that projects from the cell surface, helping the traffic of the Hh pathway proteins [12]. Disruption of midline patterning, as seen in HPE, results in the failure of prosencephalon cleavage into distinct right and left hemispheres. Deep brain structures, olfactory and optic bulbs and tracts can also be affected [3–6]. (…).

Reviewer 2 Report

The subject of the article is a topic that has been discussed in the literature in every aspect. I don't think it also has a contribution to the literature.

Author Response

Thank you for your comment. The review topic of “Holoprosencephaly” was requested by the journal.

Reviewer 3 Report

Dear authors,

Your manuscript is written in a textbook's chapter style and seems to be informative. it is well structured and may constitute an important summary in medical education.

Regarding the section on etiology, table 2 is not well organized since you dichotomized genetic and environmental (non-genetic). Please revise that table in 2 columns, and please stick to one terminology or both together (either environmental or non-genetic, or both, but not interchanging). In addition to the environmental risk factors, you can add also heavy metals and radiation exposure, although no causality has been clearly established.

The section on management is written essentially on the investigations. You may add some additional legal aspects like termination of the pregnancy when survival risk is poor, management of associated syndromes, and additional counseling of parents. 

Thank you

Author Response

We would like to thank you for your valuable and constructive comments. Please find below a point-by-point response to all the comments.

  1. Your manuscript is written in a textbook's chapter style and seems to be informative. It is well structured and may constitute an important summary in medical education. Regarding the section on etiology, table 2 is not well organized since you dichotomized genetic and environmental (non-genetic). Please revise that table in 2 columns, and please stick to one terminology or both together (either environmental or non-genetic, or both, but not interchanging).
  • Thank you very much for your comment. We have reviewed the terminology and we divided genetic causes and non-genetic risk factors into two different tables to make it clearer.
  1. In addition to the environmental risk factors, you can add also heavy metals and radiation exposure, although no causality has been clearly established.
  • We have added a discussion on heavy metals and radiation exposure, as suggested (10; l. 368-375):

(…) Exposure to heavy metals and radiation were also suggested to have deleterious effects on brain development, although no causality has been clearly established with HPE.  For example, the occupational exposure to X-rays was correlated with an increased severity of HPE cases [72], even though the maternal exposure to radiation was not associated with a higher prevalence [72,80,81]. Similarly, an African case series suggested that teratogenicity related to mining-related pollution could be responsible for the emergence of three cases of HPE [82], although the potential connection between metal exposure and HPE occurrence has not been yet properly investigated (…).

  1. The section on management is written essentially on the investigations. You may add some additional legal aspects like termination of the pregnancy when survival risk is poor, management of associated syndromes, and additional counseling of parents. 
  • Two new sections have now been added: one regarding management and another about genetic counselling ( 12; l. 452-479 and p. 11-12; l. 420-449):

(…) In the event of a prenatal diagnosis of HPE on ultrasound or fetal MRI, genetic testing can also be performed during pregnancy for the identification of chromosomal abnormalities or monogenic disorders associated with HPE. Fetal karyotype, CMA analyses or even multigene panels are possible through analysis of amniotic fluid [6].

By the time a prenatal diagnosis of HPE is typically made, the pregnancy has usually reached 18 to 22 weeks [89], facing the parents with very challenging decisions within a narrow timeframe, such as the termination of pregnancy and the establishment of postnatal goals of care for the unborn child [90].

A study conducted with mothers who received a prenatal diagnosis of alobar HPE showed that parents’ needs for information, emotional support, and assistance are often not being adequately addressed by the healthcare providers [90]. Usually, parents are given a very negative prognosis and they often expect that their children will not survive beyond a few days or weeks after birth [91]. However, the prognosis of these patients is most commonly uncertain, especially for those with less severe forms.

Recommended practices for managing these cases include: the avoidance of terms as “not viable”, “incompatible with life”, and “vegetable”; provision of balanced information on the possible outcomes (including the chance of survival beyond the perinatal period) based on both literature and other parental experiences; and referral to a specialist [90].

Meeting with a genetic counselor or a geneticist is also recommended to assist in the estimation of recurrence risk in future pregnancies [90]. Overall, it has been estimated that the recurrence risk could be up to 50% in the case of a parent carrying a mutation with incomplete penetrance and 1% in the case of aneuploidy such as trisomy 13 [30].

In the cases where a clinical diagnosis is established, but the genetic cause is not known, is not possible to predict the recurrence risk for future pregnancies [89,92]. That imposes a lot of stress and a sense of uncertainty to the parents, which face an additional hassle regarding familial planning decisions (…).

(…) It is highly recommended that the care of patients with HPE be based on a multidisciplinary approach, with involvement of gastroenterologists, neurologists, neurosurgeons, and plastic surgeons, among other health care professionals.

A detailed ophthalmologic evaluation and hearing assessment should be performed in all patients. Feeding and neurodevelopment should be closely followed. Complications related to oromotor dysfunction, aspiration, endocrinologic dysfunction, hydrocephalus and seizures are frequent and should be monitored [22].

Pharmacological treatment of gastroesophageal reflux may be necessary, and, in severe cases, Nissen fundoplication or transpyloric feed can be recommended [30]. When indicated, gastrostomy tube placement might help prevent respiratory complications and reduce hospitalization rates.

Evaluation by a pediatric endocrinologist should be performed on a regular basis, given the common occurrence of endocrinologic dysfunction [2].

Movement disorders and prominent spasticity with possible progression to orthopedic deformities are often present and may become more prominent during the follow-up period. The management of these conditions might resemble the patients with cerebral palsy and the inclusion of physical and occupational therapy represent good preventive measures [30]. Cases with refractory pain or functional limitations might benefit from intramuscular botulin toxin injections, oral antispasmodic agents could or even orthopedic surgery for those with complications, such as contractures, hip dislocation and scoliosis [30].

Challenges and needs of the patients evolve over their lifespan. Individuals with HPE who are higher functioning may experience neuropsychologic problems, including executive dysfunction, attention deficit hyperactivity disorder, and learning disabilities [30]. Psychologic/psychiatric intervention is frequently the first line of approach recommended, but upon the eventual emergence of complications (i.e. anxiety and depression) pharmacologic treatment may be necessary.(…).